# Landscape structure evolution and ecological risk evaluation of oasis desert cities: A case study of Tiemenguan city

Mingyue Sun[1], Hongguang Liu[1]*, Yingsheng Dang[1]*, Ping Gong[1], Pengfei Li[1], Rui Fang[2], Huan Cao[2], Xiang Li[2], Hanji Xia[1], Fuhai Ye[1], Yong Guo[1]

1 College of Water Conservancy and Architectural Engineering, Shihezi University, Shihezi, Xinjiang Province, China, 2 Hydrology and Water Resources Management Center of the Second Division of Xinjiang Production and Construction Corps, Tiemenguan, Xinjiang Province, China

* liuhongguang-521@163.com (LH); 1075951141@qq.com (DY)

## Abstract

The rapid development of oasis desert cities adversely affects fragile ecosystems, preventing regional sustainable development. This study investigates the spatio-temporal evolution characteristics and potential quantitative relationship between oasis landscape structure (OLS) and the ecological risk index (ERI) and the trend in different development scenarios in Tiemenguan City, a typical oasis city in an arid zone in northwestern China, from 1990 to 2020. We calculated the ERI thresholds for different landscape types, classified ecological risk levels, and examined the factors influencing ecological risk. The normalized difference vegetation index (NDVI) thresholds were NDVI ≥ 30% for oases, 10% <NDVI < 30% for transition zones, and NDVI ≤ 10% for desert areas. Under government control, transitions from cropland, woodland, and grassland to built-up and unused land decreased by 20%, whereas conversions from unused land to cropland, woodland, and grassland increased by 30%. The results showed the following: (1) The oasis area expanded continuously from 175.5 km² to 345.3 km² during 30 years. The transition and desert zones decreased by 49.7% and 37.9%, respectively. The ERI decreased and was strongly correlated with the OLS. The thresholds of the ERI in the oasis zone-transition zone and the transition zone-desert zone were 0.08–0.085 and 0.111–0.118, respectively. (2) Socioeconomic factors, including infrastructure expansion, population density, and GDP, were dominant influences, contributing 64% to the ERI, whereas the influence of natural factors such as climate declined. (3) The low-ERI areas increased by 3.3% under government control, and the transition zones increased significantly, slowing the growth rate of the oasis zone. This study quantitatively evaluated the landscape types' ecological risk levels and analyzed the effects of dynamic migration on the landscape type stability. This paper provides a systematic research framework for ecological risk assessment of various landscape types in oasis desert cities and a scientific basis for ecological conservation and related research.

**Data availability statement:** All relevant data are within the manuscript and its Supporting Information files.

**Funding:** This research study was supported by both Divisional and Municipal Science and Technology Tackling Program Projects (2024GG2301 and 2023GG1502). The role of the funder in the research is as follows: Prof. Liu Hongguang and Mr. Dang Yingsheng are the leaders of Projects (2024GG2301 and 2023GG1502), respectively, and their respective contributions to the study are listed below: Hongguang Liu: Conceptualization, Data curation, Funding acquisition, Methodology, Project administration, Resources, Writing – review & editing Yingsheng Dang: Formal analysis, Investigation, Funding acquisition, Project administration, Supervision, Validation, Writing – review & editing We have also added an acknowledgment of the division's science and technology programs (2024GG2301 and 2023GG1502) in the "Acknowledgements" section on line 637-642 of the revised version. We thank Divisional and Municipal Science and Technology Tackling Program Projects (2024GG2301 and 2023GG1502)] for the support of this study in terms of funding as well as article writing.

**Competing interests:** The authors have declared that no competing interests exist.

## Introduction

Oases are crucial in ecological conservation and agricultural production in arid zones. They typically cover less than 10 percent of the region's area but support more than 90 percent of the population [1]. The development of oases depends on water and heat conditions. These areas generally have an ecological gradient, i.e., the vegetation cover, biodiversity, and other characteristics change significantly from the desert to the oasis areas. The vegetation zone separating the desert zone from the oasis zone represents a transitional buffer (i.e., it captures wind-driven sand, serves as a soil and water conservation area, and ensures species diversity). As a result, the landscape is comprised of an oasis area, a transition zone, and a desert area from the inside to the outside [2]. Various factors, such as climate change and human activities, influence the evolution of the landscape structure. Rational land development and water management have accelerated the expansion of oasis zones [3]. However, wind erosion and water shortages adversely affect newly reclaimed areas, reducing the region's ecological stability, preventing oasis expansion, and potentially causing ecosystem degradation and desertification in local areas [4].

Analyzing changes in the oasis landscape structure (OLS) is significant for maintaining regional ecological stability and sustainable socioeconomic development. Consequently, it is a hot research topic in arid and semi-arid regions [5]. Spectral index thresholding has been widely applied. The normalized difference vegetation index (NDVI) is commonly used to detect differences in vegetation cover over time and analyze landscape change. However, the thresholds depend on the region and the remote sensing data source. For example, Ding et al. used NDVI values to delineate different zones. They defined areas with NDVI values of less than 15% as desert zones, those with values from 15%-30% as transition zones, and those with NDVI values ≥ 30% as oasis zones [6]. A study of oasis cities in the middle part of the Tarim River found that the average vegetation cover in the oasis zone-transition zone was 24% [7]. Similarly, Zhang et al. set the NDVI thresholds at 20% (oasis zone-transition zone) and 35% (transition zone-desert zone). A study of oasis desert zones in the Hexi Corridor obtained an overall accuracy (OA) of the delineation results of more than 80% [8]. Therefore, choosing NDVI thresholds suitable for regional characteristics is critical to ensure high accuracy in classifying oasis landscape zones.

Ecological risk refers to potential ecosystem damage, such as land degradation, landscape fragmentation, and biodiversity loss [9–11]. Multiple landscape ecosystems, such as mountains [12], lochs [13], oases [14], and cities [15], have been analyzed. The geographical characteristics of the oasis region, such as fragile ecosystems and low resilience, have increased the difficulty and urgency of ecological risk management in the region. Many studies have used multiple ecological risk indicators (ERIs), including the ecological carrying capacity index [16], the ecological vulnerability index [17], the landscape disturbance index (LDI) [18], and the degree of landscape ecological loss [19], to establish comprehensive and precise ecological risk evaluation frameworks. Due to the complex mechanisms of natural and

socioeconomic factors and difficulties in data acquisition, many methods have been used to analyze complex interactions between factors and ERIs, such as geographically weighted regression [20], Random Forest (RF) regression [21], structural equation modeling [22], and spatial error modeling [8]. Thus, it is crucial to incorporate regional characteristics and use multiple indicators and advanced analytical methods in analyzing the ecological risk of oasis areas.

Simulating different future scenarios is necessary to evaluate risk. Model simulations have been used to predict future development and the dynamic evolution of oases. The Future Land Use Simulation (FLUS) model [23], Patch-generating Land Use Simulation (PLUS) model [24], Conversion of Land Use and its Effects at Small regional extent (CLUE-S) model [25], and the Cellular Automata (CA)-Markov model [26] have been widely used. The applicability of a model depends on its ability to deal with spatial heterogeneity, data requirements, and ecosystem complexity. Suitable models must be chosen based on the research objectives and data, and their strengths and weaknesses should be considered to obtain an accurate prediction of the development of oasis regions.

Xinjiang is located in an arid region. It faces unique ecological and socioeconomic challenges. Tiemenguan City, a representative oasis city in Xinjiang, has achieved significant economic development due to initiatives like the Western Development Program. However, the city has high water scarcity and an unstable ecosystem, adversely impacting environmental sustainability and human well-being [27,28]. This study examines the correlation between the OLS and ERI in oasis areas. We assess the spatiotemporal characteristics of OLS and ERI and the impact of dynamic migration in the oasis area on the ecological stability of the transition zone and the desert area to determine whether the degradation of the transition zone increases the ecological risk of the oasis area. An ERI quantifies the potential risks associated with changes in landscape patterns to provide risk warnings for urban development. The objectives of this study are to (1) analyze the potential correlation between the spatiotemporal characteristics of OLS and ERI from 1990 to 2020, (2) assess the relative importance of drivers of spatiotemporal changes using the ERI and the RF regression model, and (3) predict trends in the ERI and OLS for different land use scenarios from 2020 to 2030 using the PLUS model. Fig 1 shows the research framework.

## Materials and methods

### Study area

Tiemenguan City is located at the intersection of the southern foothills of the Tianshan Mountains and the northern slopes of the Kunlun Mountains (Fig 2). It is located on an important route in the middle section of the ancient Silk Road (41°45′–41°49′N, 85°53′–86°05′E). The area is located on the Eurasian continent, far from the sea. It has a warm-temperate continental arid climate, with arid and hot summers and cold winters. The average annual rainfall is 100–200 mm, most of which falls in the summer, making it a highly arid zone. Cultivated and unutilized land account for the largest proportions of land use types in the study area (Standard for Classification of Current Land Use Situations (GBT21010–2007)). The terrain is high in the north and low in the south. Due to the Western development strategy, the population increased from $1.88 \times 10^4$ in 1990 to $2.39 \times 10^4$ in 2020, and the GDP grew from $4.62 \times 10^8$ CNY to $1.85 \times 10^{10}$ CNY. Therefore, the study area's ecological pattern underwent a significant change. The expansion of cultivated land and the development of unutilized land significantly contributed to regional economic growth. However, rapid development has also put tremendous pressure on fragile ecosystems.

### Data sources

We used three data sources:

(1) Remote sensing images: We used Landsat series images from 1990 to 2020 (with an interval of 10 years). We selected images from June to September with less than 10% cloud cover. The NDVI was used to classify oasis, transition, and desert zones. Misclassifications were corrected using land-use data and high-resolution satellite images (S1

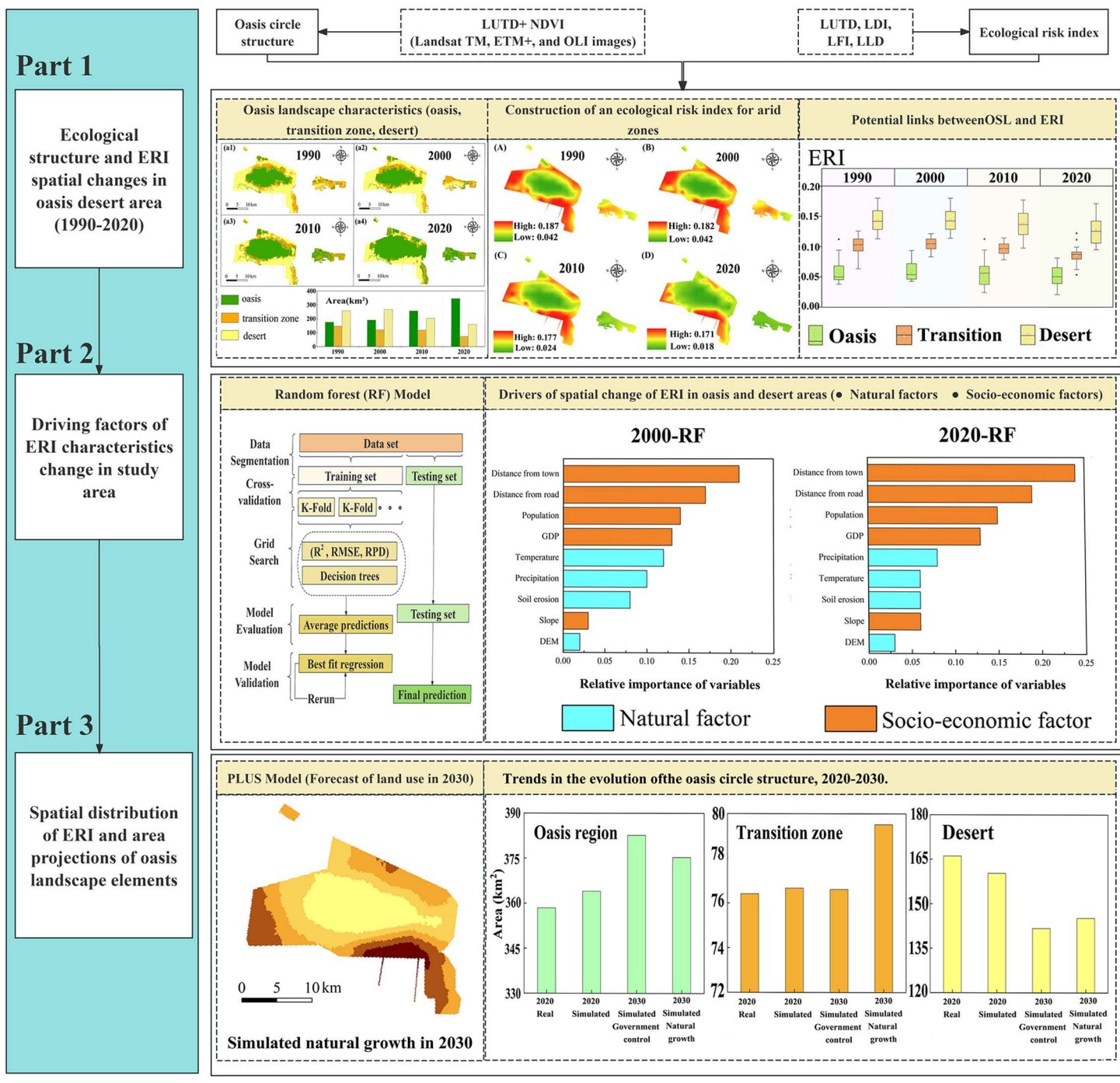

**Fig 1. Research framework.** Notes: LUTD, LDI, LFI, and LLD stand for Land Use Type Data, Landscape Disturbance Index, Landscape Fragility Index, and Landscape Loss Index, respectively.

Fig). The steps were as follows: (a) ENVI 5.3 was used to pre-process the images using splicing, cropping, radiometric calibration, and atmospheric correction. The NDVI of the images from 4 dates (resolution of 30 × 30m) was calculated. (b) Google Earth high-resolution images were used to determine the NDVI thresholds between the oasis, transition,

and desert zones (NDVI ≥ 30% for the oasis zone, 10 <NDVI < 30% for the transition zone, and NDVI ≤ 10% for the desert zone). (c) The classification errors were corrected using visual interpretation and land use data. (d) The accuracy of the classification results was determined by selecting 2000 random points and using Google Earth high-resolution images. The OA and Kappa coefficient were above 80%, meeting the research requirements (Table 1).

(2) Land use data: Land use data were obtained from the Center for Resource and Environmental Science and Data of the Chinese Academy of Sciences (www.resdc.cn).

(3) Geographic data: We used GDP, population, elevation (digital elevation model (DEM), slope, soil erosion, average annual temperature, annual rainfall, and distance from roads and towns. The distances were extracted using Euclidean distance analysis in ArcGIS 10.8. Table 2 lists the data sources.

## Calculation of ERI

An ecological risk assessment identifies and quantifies potential risk factors [11]. The ERI has been widely used in ecological risk assessments and studies of changes in the landscape structure [29]. We analyzed the spatial distribution and evolution of different landscape types in the region to assess the risk to oasis urban ecosystems. The landscape

**Table 1. Classification accuracy.**

| Year | | Oasis | Transition zone | Desert | UA | PA | OA | Kappa |
|---|---|---|---|---|---|---|---|---|
| 1990 | Oasis | 536 | 66 | 3 | 0.91 | 0.89 | 0.90 | 0.84 |
| | Transition zone | 34 | 436 | 54 | 0.81 | 0.83 | | |
| | Desert | 16 | 33 | 822 | 0.94 | 0.94 | | |
| 2000 | Oasis | 566 | 41 | 7 | 0.88 | 0.92 | 0.91 | 0.86 |
| | Transition zone | 49 | 388 | 30 | 0.85 | 0.83 | | |
| | Desert | 29 | 29 | 861 | 0.96 | 0.94 | | |
| 2010 | Oasis | 748 | 21 | 15 | 0.93 | 0.95 | 0.91 | 0.86 |
| | Transition zone | 43 | 382 | 40 | 0.86 | 0.82 | | |
| | Desert | 16 | 41 | 694 | 0.93 | 0.92 | | |
| 2020 | Oasis | 1024 | 26 | 23 | 0.97 | 0.95 | 0.92 | 0.86 |
| | Transition zone | 23 | 273 | 44 | 0.82 | 0.80 | | |
| | Desert | 9 | 35 | 514 | 0.88 | 0.92 | | |

**Note:** OA: overall accuracy; Kappa: kappa coefficient; UA: user accuracy; PA: producer accuracy.

**Table 2. Geographic data.**

| Data usage | Category | Data index | Data Resource |
|---|---|---|---|
| Assessment of ERIs | Natural factors | Elevation | Geospatial Data Cloud (https://www.gscloud.cn/) |
| | | Slope | |
| | | Temperature | National Centre for Atmospheric Science (https://ncas ac.uk/) |
| | | Precipitation | |
| | | Soil erosion | Resource and Environmental Science and Data Center (https://www.resdc.cn) |
| | Socioeconomic factors | GDP | |
| | | Population | |
| | | Distance from roads | OpenStreetMap (https://www.openhistoricalmap.org) |
| | | Distance from towns | |

fragmentation index ($C_i$), landscape separation index ($S_i$), landscape dominance index ($D_i$), Landscape Disturbance Index (LDI), and Landscape Fragility Index (LFI) were used to establish the ERI [30]. The $C_i$ was chosen because land development in the study area led to increased fragmentation, reducing ecosystem stability and increasing the risk of habitat loss. The $S_i$ was used to reflect the connectivity of landscape patches, which is critical in oases with frequent human activities. The $D_i$ reflects dominant landscape types based on the $S_i$. It was used to determine the influence of dominant landscape types, such as unused land and cultivated land, on ecosystems and their trends. The $D_i$ incorporates fragmentation, separation, and dominance and enables the quantification of disturbances of oases in arid zones affected by human activities. The LFI is an indicator a landscape's disturbance resistance and recovery ability. It reflects the impact of the vulnerability of unutilized land and grassland on ecosystem stability [31]. These indicators accurately reflect the relationship between landscape patterns and ecological risk. A 1 km × 1 km grid (Fig 2) was created in ArcMap 10.8 to delineate the study area. Spatial interpolation was used to determine the risk in each grid. The equations for calculating the indicators are listed in Table 3.

We assigned different weights to utilized and unutilized land to calculate the LDI. The weights of $C_i$, $S_i$, and $D_i$ were 0.5, 0.3, and 0.2 in utilized land and 0.3, 0.2, and 0.5 in unutilized land, respectively. The weights were chosen based on the different indicators of ecosystem stability and risk assessment. The study area is an oasis with different land use types. Fragmentation of utilized land significantly increases habitat loss and ecosystem vulnerability; thus, we used a higher weight (0.5). In contrast, the impact of fragmentation on ecosystem stability is lower in unutilized land; therefore, we used a lower weight (0.3). The $S_i$ was used to measure the connectivity of landscape patches. All landscape types were assigned medium or low weights (0.3 and 0.2) because of the relatively low effects of this indicator on ecosystem stability. The $D_i$ reflects the dominance of landscape types. A higher weight was used for unutilized land (0.5), and a lower weight was used for utilized land (0.2).

This study used a spatial autocorrelation model (via GeoDa software) to analyze the global and local autocorrelation of ecological risk and reveal the spatial distribution characteristics of the ERI [32]. Spatial heterogeneity and similarity were quantified using the global Moran's Index. Values close to 1 indicate strong positive spatial autocorrelations between the ecological risk eigenvalues, values close to -1 indicate a negative correlation, and a value of 0 indicates no significant spatial autocorrelation [33]. Spatial clustering of high and low ERI values based on the global Moran's Index was used to classify the study area into four types: High-High, High-Low, Low-High, and Low-Low to reveal the spatial aggregation and dispersion patterns of ecological risks. The equations are as follows [34]:

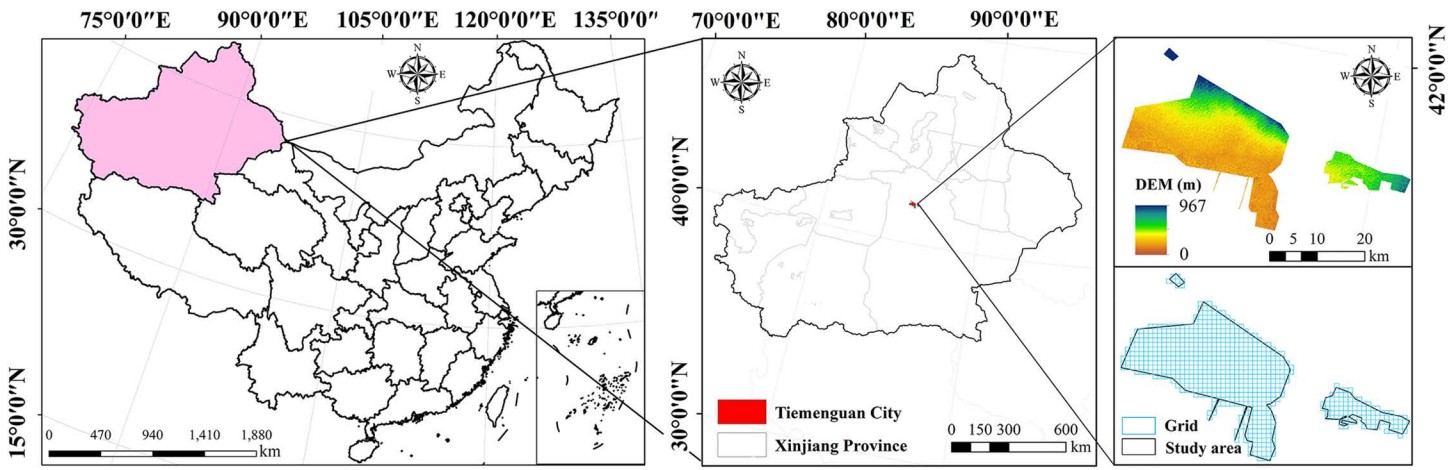

**Fig 2. Location of the study area.**

$$GlobalMoran's\ Index = \frac{n \sum_{i=1}^{n} \sum_{j=1}^{n} w_{ij}\left(x_i - \overline{x}\right)\left(x_j - \overline{x}\right)}{\left(\sum_{i=1}^{n} \sum_{j=1}^{n} w_{ij}\right) \sum_{i=1}^{n}\left(x_i - \overline{x}\right)^2} \tag{1}$$

$$LocalMoran's\ Index = \frac{n\left(x_i - \overline{x}\right) \sum_{j=1}^{n} w_{ij}\left(x_j - \overline{x}\right)}{\sum_{i=1}^{n}\left(x_i - \overline{x}\right)^2} \tag{2}$$

where $n$ is the number of samples, $x_i$ and $x_j$ denote the attribute values of $i$ and $j$, respectively, $\overline{x}$ is the mean value, and $w_{ij}$ is the spatial weight matrix.

## Driving mechanisms of ecological risk

This study investigated the driving mechanism of ecological risk using an RF regression model (shown in S2 Fig) [34]. RF is an ensemble algorithm with a better learning performance than a single decision tree model. The model determines the importance of the covariates. This method has been widely used in many fields, such as data mining and ecology. The algorithm uses bootstrap sampling, where N training sets are sampled with replacement (about 2/3 of the total number of samples), and the results are evaluated using the remaining samples as a test set. Multiple decision trees are generated using the training set, and m of the M feature variables are randomly selected as node divisions. The prediction results of the N decision trees are used to determine new sample categories by voting. An RF model with N decision trees is constructed and scored by the test set, and the variable importance is calculated. The feature importance score (IS) is calculated as follows:

$$IS = \frac{\sum\left(errOOB2 - errOOB1\right)}{N} \tag{3}$$

**Table 3. Equations for calculating the ecological risk indicators.**

| Name | Descriptive | Formulas | Description of the formula |
|---|---|---|---|
| Landscape fragmentation index ($C_i$) | The degree of patch fragmentation of a landscape type | $C_i = \frac{N_i}{A_i}$ | $N_i$ represents the number of patches in landscape $i$; $A_i$ represents the area of landscape $i$; A is the total area. |
| Landscape separation index ($S_i$) | The degree of separation of patches in a landscape type | $S_i = \frac{A}{2A_i}\sqrt{\frac{N_i}{A}}$ | |
| Landscape dominance index ($D_i$) | The degree of importance of patches in a landscape type | $D_i = \frac{Q_i + M_i}{4} + \frac{L_i}{2}$ | $Q_i$ represents the ratio of the number of samples to the total number of samples in landscape $i$; $M_i$ represents the ratio of the number of patches to the total number of patches in landscape $i$; $L_i$ represents the ratio of the area of the number of patches to the total area in landscape $i$. |
| Landscape disturbance index (LDI) | The degree of disturbance of a landscape type | $LDI = a \times C_i + b \times S_i + c \times D_i$ | a, b, and c are the weights of the landscape indicators. Their values were based on experience: a = 0.5, b = 0.3, and c = 0.2 in utilized land (in unutilized land, the weights were 0.3, 0.2, and 0.5, respectively [14]. |
| Landscape fragility index (LFI) | The resistance of a landscape type to disturbance | The values were manually assigned based on previous studies and were modified according to study area conditions | The six landscape types were assigned vulnerability values from high to low (construction land (6), forest land (5), arable land (4), grassland (3), water (2), and desert (1)). The data were normalized to obtain the vulnerabilities of different landscape types [14]. |
| Ecological risk index (ERI) | Combined ecological risk level for all samples | $ERI_k = \sum_{i=1}^{n} \frac{A_{ki}}{A_k} LDI \cdot LFI$ | $ERI_k$ is the ecological risk index of the kth sampling area, $A_{ki}$ is the area of landscape type i in the kth sampling area, and $A_k$ is the total area of the kth sampling area. |

Each decision tree's out-of-bag data *error (errOOB1)* is computed, and *errOOB2* is the error after random noise interference is added to all sample features of the out-of-bag data.

The model parameters were screened using a grid search and 4-fold cross-validation to improve model performance [35]. The parameter ranges were as follows: n_estimators ranged from 5–50 in steps of 1, max_depth ranged from 1–20 in steps of 1, max_features ranged from 1–10 in steps of 1, and the random_state was 1. The process was implemented in MATLAB R2023a.

The parameters used for model evaluation were the determination coefficient ($R^2$), root mean square error (RMSE), and relative percent deviation (RPD). When RPD ≥ 2, The model performance is excellent when RPD ≥ 2, fair when 1.4 ≤ RPD < 2, and low when RPD < 1.4 [36].

$$R^2 = \frac{\sum_{i=1}^{n} (\hat{y}_i - \bar{y}_i)^2}{\sum_{i=1}^{n} (y_i - \bar{y}_i)^2}$$

(4)

$$RMSE = \sqrt{\frac{\sum_{i=1}^{n} (\hat{y}_i - y_i)^2}{n}}$$

(5)

$$RPD = \sqrt{\frac{\sum_{i=1}^{n} (y_i - \bar{y})^2}{\sum_{i=1}^{n} (\hat{y}_i - y_i)^2}}$$

(6)

where $y$ is the measured value of ERI, $\hat{y}_i$ is the predicted value of ERI, $\bar{y}_i$ is the mean value of ERI content, and n is the total number of samples ($i = 1, 2, 3...n$).

We conducted a quantitative analysis of the contribution of ecological risk as a driving mechanism. The ERI values in 2000, 2010, and 2020 (1990 was not analyzed due to incomplete data) were used as the dependent variable. Elevation, Slope, Soil erosion, Temperature, Precipitation, GDP, Population, Distance from roads, and Distance from towns were used as independent variables. The 646 sample points were randomly divided into a training set (484) and a validation set (162). The original data were linearly normalized to ensure model stability.

**Predicting changes in land use types, OLS, and ERI**

We used the PLUS model (version 1.3.5) to simulate future land use change scenarios. The model combines meta-CA and patch generation simulation and contains modules for land expansion extraction, land expansion analysis strategy (LEAS), demand forecasting, and scenario diversity. The LEAS extracts the land use changes in two periods, analyzes the driving factors using the RF algorithm, and obtains the development probability of each type. The number of regression trees was 40, and the sampling rate and number of RF features were 0.01 and 12, respectively. The PLUS model dynamically simulates the generation of patches under the constraint of development probability [37,38].

Before the simulation, the Markov chain was used to predict land changes, and the results were used as model parameters. The land use changes in the study area in 2020 were simulated based on the land use data in 2000 and 2010. We compared the results with actual conditions in 2020 and obtained a Kappa coefficient of 0.90 and an OA of 0.94, indicating the model's reliability for subsequent analyses.

We simulated the land use change patterns under a natural growth scenario and a government regulation scenario and analyzed the spatial distribution characteristics of ERI and the quantitative characteristics of OLS under different scenarios [14,34]. In the natural development scenario, we assume that the oasis desert region would continue to exist without significant intervention. We expected the expansion of cultivated land and urban areas to continue at the current rate and desertification to proceed in areas with minimal ecological management. The natural scenario assumed that environmental factors, such as precipitation, temperature, and soil erosion, would follow historical trends, and the primary drivers of

change were natural climate variability and land-use practices. The land use change was simulated using the Markov chain model (no change). In the government-controlled scenario, the parameters in the Markov chain model reflect policy interventions aligned with ongoing government efforts to promote sustainable development and ecological restoration. We assumed a 20% reduction in the transfer of cropland, woodland, and grassland to built-up and unused land to reflect the stricter land-use regulations under the "Green Development Policy," which aims to curtail the expansion of construction projects in ecologically sensitive areas. This strategy is based on the National Territorial Planning Program (https://www.gov.cn/gongbao/content/2017/content_5171326.htm). The 30% increase in the conversion of unused land to cropland, woodland, and grassland is based on the government's large-scale ecological restoration programs, such as afforestation and grassland restoration projects designed to improve land productivity and mitigate desertification. This strategy is based on the "14th Five-Year Plan" for ecological environmental protection in Xinjiang (https://sthjt.xinjiang.gov.cn/xjepd/zhywchjgh/202205/a04450440cc5411da3d943f6668d9c3a.shtml). The 10% increase in the conversion of built-up land to woodland and grassland aligns with policies promoting green urban infrastructure and ecological zoning, emphasizing the transformation of underutilized urban areas into green spaces to enhance urban sustainability and resilience(https://www.xinjiang.gov.cn/xinjiang/gtkjgh/202407/192ba3b576af4747bf5a30fc2a520fa3.shtml). These policy-driven interventions aim to balance ecological conservation with sustainable urban development and ecological stability.

## Results

### Spatial and temporal evolution of OLS and ERI, 1990–2020

**Spatial and temporal evolution of OLS.** From 1990 to 2020, the oasis area expanded from the core area to the outside, significantly reducing desert areas in all directions. Over time, many oasis patches expanded, connected, and formed a large contiguous oasis belt. The trend exhibited stages with small and large rates of change (Fig 3a, Table 4). The period from 1990 to 2000 (the first stage) showed a small rate of change in the oasis zone (about 8.5% growth) and a trend of expansion in the desert zone (about 4.3% expansion), with limited spatial expansion of the elements. The change rate in the landscape structure increased significantly, with the oasis zone expanding from 190.6 km$^2$ to 345.3 km$^2$ and the transition and desert zones declining significantly by 38.7% and 40.3%, respectively.

The differences in spatial transitions between the oasis, transition, and desert zones were evident from 1990 to 2020 (Fig 3b). In the first stage, the conversion from desert to oasis was not significant (changes in all factors were below 38 km$^2$), indicating that the effectiveness of ecological management was limited during this period, and the land degradation trend was not significantly curbed. Conversely, the second phase was a critical period of oasis expansion, with oases acquiring a significant area from transition zones and deserts (81.67 and 36.4 km$^2$, respectively). The desert area decreased significantly from 2010 to 2020 and was transformed into transition and oases zones, with areas of 41.02 and 31.38 km$^2$, respectively.

The region's ecological environment improved significantly from 1990 to 2020, and especially after 2000. The oasis area expanded, and deserts were transformed into oasis and transition zones. This trend shows that land degradation was controlled by effective ecological management measures during the past 30 years.

### Spatial and temporal evolution of land-use type and transfer

A comprehensive analysis of 30 years of land use data and its changes (Table 5) shows that the study area has a high percentage of cropland, medium to high-cover grassland, and unused land (more than 90% of the total). Due to the implementation of the Western Development Strategy, the land use change from 2000 to 2020 differed significantly from that of the previous decade (Fig 4a). From 1990 to 2020, cropland and construction land increased by 40.84% and 342.29%, and forest land and grassland decreased by 37.60% and 53.96%, respectively. More land use change occurred in the southern and eastern regions of the study area, where the production capacity of arable land declined, and land degradation was not effectively curbed. Fig 4b depicts the magnitude and direction of land-use change over time, indicating substantial cross-regional shifts between unutilized land and grassland.

**Spatial and temporal evolution of ERI.** The spatial distribution and trend of the ERI and Moran's Index from 1990 to 2020 (Fig 5) indicate a significant reduction in high ecological risk areas, especially in the eastern part of the study area, suggesting that rational land planning has improved ecological conditions. Low ERI areas increased over time, indicating a broad reduction in ecological risk and a transition from high-risk to low-risk areas in multiple locations. This improvement likely resulted from targeted interventions, such as land management reforms. The ERI levels are significantly lower in the central part of the study area, indicating that this region is ecologically stable and has low risk. Ecological risk management in this region has shown favorable results after 30 years. The percentage change of areas with different ERI levels is shown in Fig 5a. The percentage of high-risk areas was 21.41% in 1990 and decreased significantly (6.8%) after 30 years. The percentage of low-risk areas increased significantly, indicating that the region's ecological risk was mitigated. The ERI exhibited variability during the 30-year period (Fig 5b). From 1990 to 2000 (Fig 5b-5I), regions with High-High ERI values were concentrated at the periphery of the study area, whereas Low-Low regions were primarily located in the center of the oasis. In contrast, the High-High areas decreased significantly in the 2010–2020 period (Fig 5b-5K), whereas the Low-Low areas expanded significantly, indicating a reduction in ecological risk. Fig 5c shows the spatial distribution of the global and local Moran's indices. Spatial clustering of ecological risks

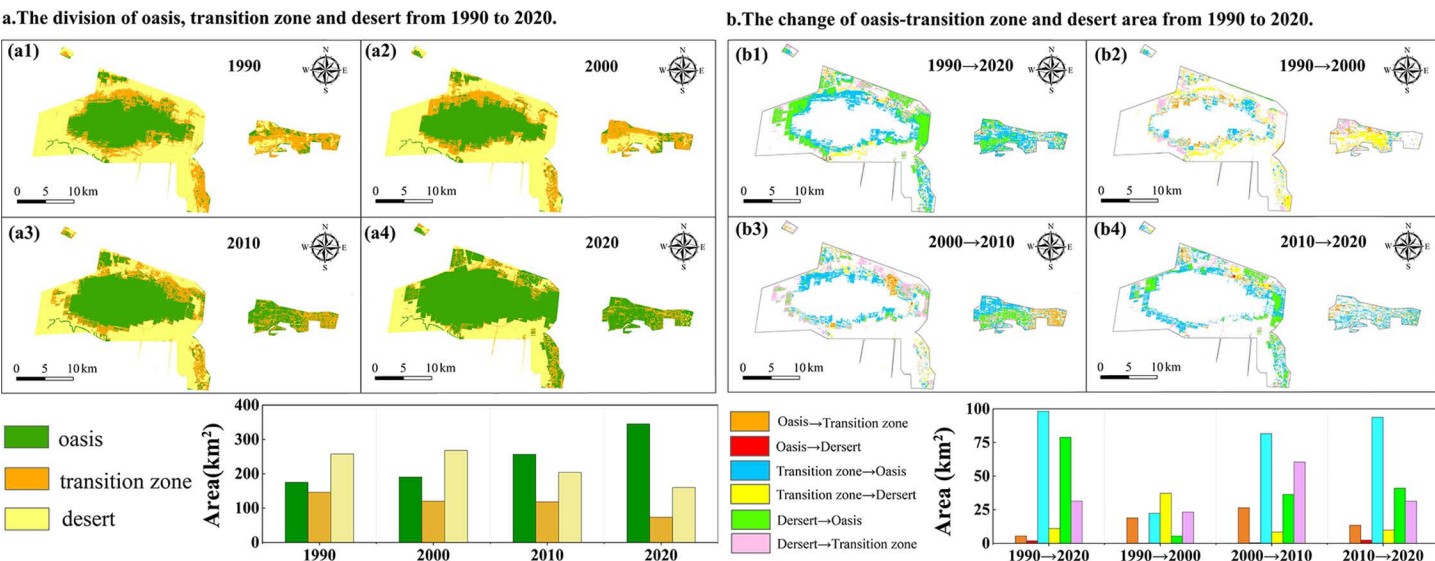

**Fig 3. Areas of oasis, transition, and desert zones and shifts in landscape patterns from 1990 to 2020.**

**Table 4. Area statistics for oases, transition, and desert zones and changes in landscape patterns from 1990 to 2020.**

| Year | Oasis | Transition zone | Desert | Oasis→ Transition zone | Oasis→ Desert | Transition zone→ Oasis | Transition zone→ Desert | Des-ert→ Oasis | Desert→ Transition zone |
|---|---|---|---|---|---|---|---|---|---|
| 1990 | 175.5 | 146.3 | 257.8 | / | / | / | / | / | / |
| 2000 | 190.6 | 120.4 | 268.6 | / | / | / | / | / | / |
| 2010 | 256.8 | 118.9 | 204.0 | / | / | / | / | / | / |
| 2020 | 345.3 | 73.6 | 160.0 | / | / | / | / | / | / |
| 1990-2020 | / | / | / | 5.54 | 1.93 | 98.25 | 11.14 | 78.94 | 31.45 |
| 1990-2000 | / | / | / | 18.96 | 0.21 | 23.22 | 37.27 | 5.44 | 23.35 |
| 2000-2010 | / | / | / | 26.51 | 0.42 | 81.66 | 8.43 | 36.39 | 60.44 |
| 2010-2020 | / | / | / | 13.49 | 2.42 | 93.74 | 9.93 | 41.02 | 31.38 |

increased during the 30-year period (the global Moran's Index increased from 0.717 to 0.792). Regions with Low-Low values in the eastern part of the study area increased sharply from 2000 to 2020, and the degree of ecological fragmentation decreased significantly. The distribution of high-risk and low-risk areas exhibited negligible change over time, and the spatial heterogeneity of risk increased. These results suggest that land-use changes and ecological management measures have mitigated ecological risks in the region. However, local clustering of high-risk areas was observed.

**Potential link between ERI and OLS.** Fig 6 illustrates the temporal dynamics of the ERI in the oasis, transition, and desert zones from 1990 to 2020. The oasis zones are characterized by dense vegetation cover, effective land management practices, and ecological restoration measures. They consistently have lower ERI values (ranging from 0.05 to 0.06), indicating they are relatively stable and resilient. The lower risk is attributed to the higher biodiversity and soil and water conservation measures [39]. In contrast, transition zones exhibit higher variability in ERI values (ranging from 0.085 to 0.102), reflecting higher sensitivity to disturbance. These zones are located at the boundary between the oasis and desert areas. They are ecologically unstable and more vulnerable to disturbance. The larger range in ERI values suggests that natural factors, such as wind erosion and water scarcity, and anthropogenic pressures, like land use changes and agricultural expansion affect the transition zones significantly [40]. In contrast, desert zones have the highest ERI values

**a. Land use spatial distribution and changes (1990–2020).**

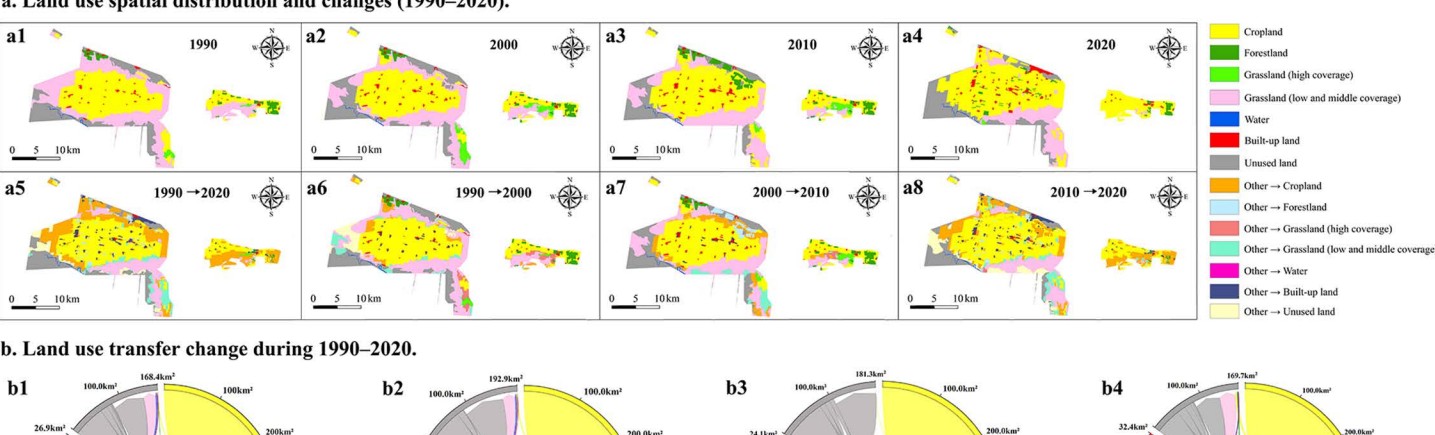

**b. Land use transfer change during 1990–2020.**

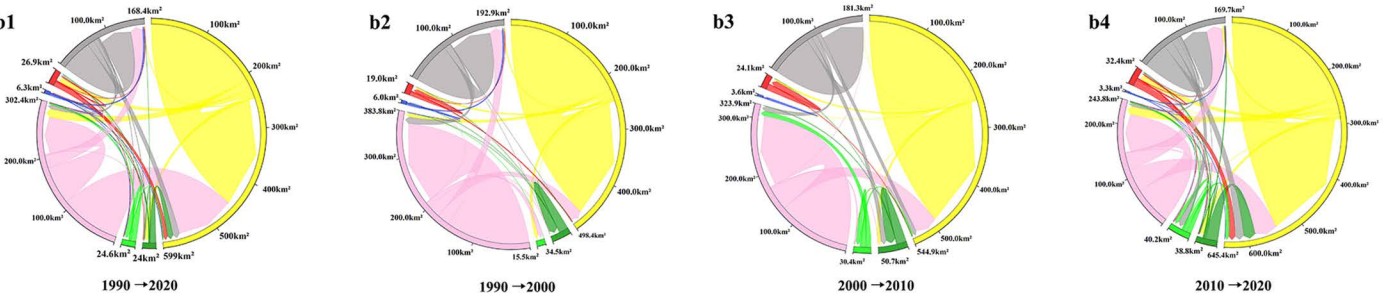

**Fig 4. Land use change from 1990 to 2020.**

**Table 5. Areas of different land use types from 1990 to 2020 (km²).**

| Year | Cropland | Forestland | Grassland (high coverage) | Grassland (low and medium coverage) | Water | Built-up land | Unused land |
|------|----------|------------|----------------------------|--------------------------------------|-------|----------------|-------------|
| 1990 | 247.48 | 16.94 | 4.43 | 209.33 | 2.57 | 8.37 | 88.75 |
| 2000 | 250.94 | 17.58 | 19.89 | 174.49 | 1.80 | 13.66 | 104.10 |
| 2010 | 294.00 | 33.13 | 10.46 | 149.41 | 1.80 | 23.47 | 77.19 |
| 2020 | 348.56 | 10.57 | 1.01 | 86.37 | 1.79 | 37.02 | 94.99 |

**a. The ERI Spatial-temporal distribution from 1990 to 2020.**

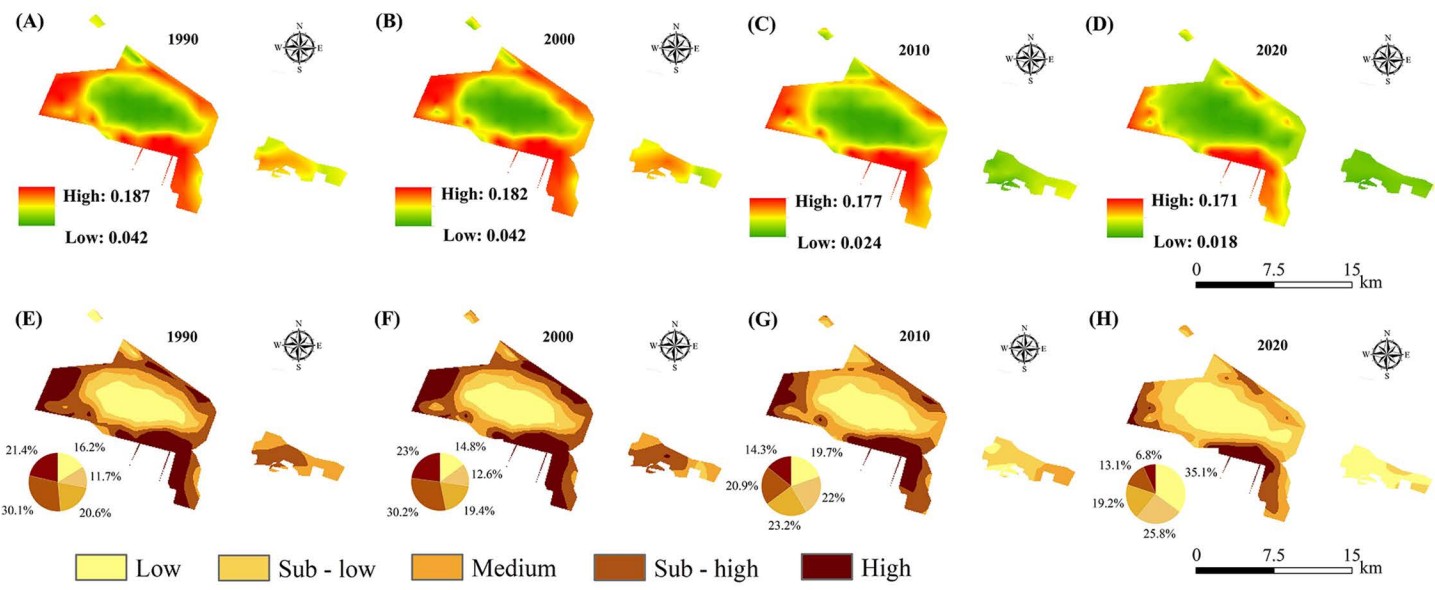

**b. The spatially variable distribution of ERI from 1990 to 2020.**

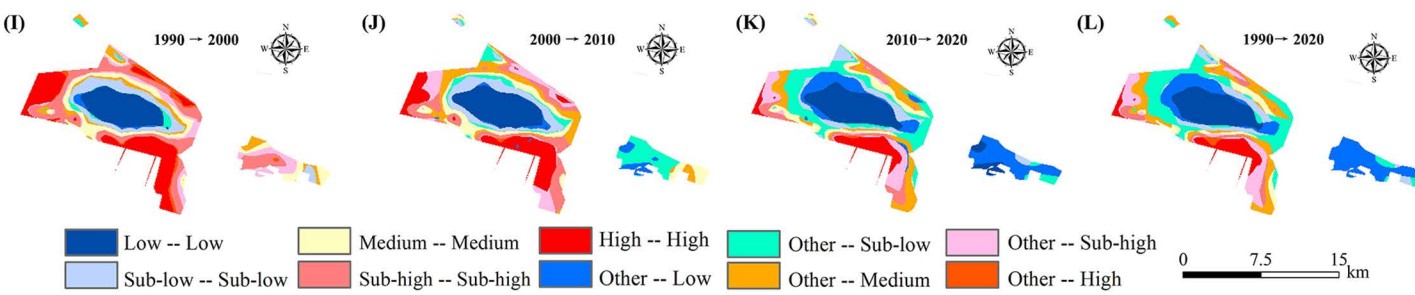

**c. The spatial distribution of Moran's Index from 1990 to 2020.**

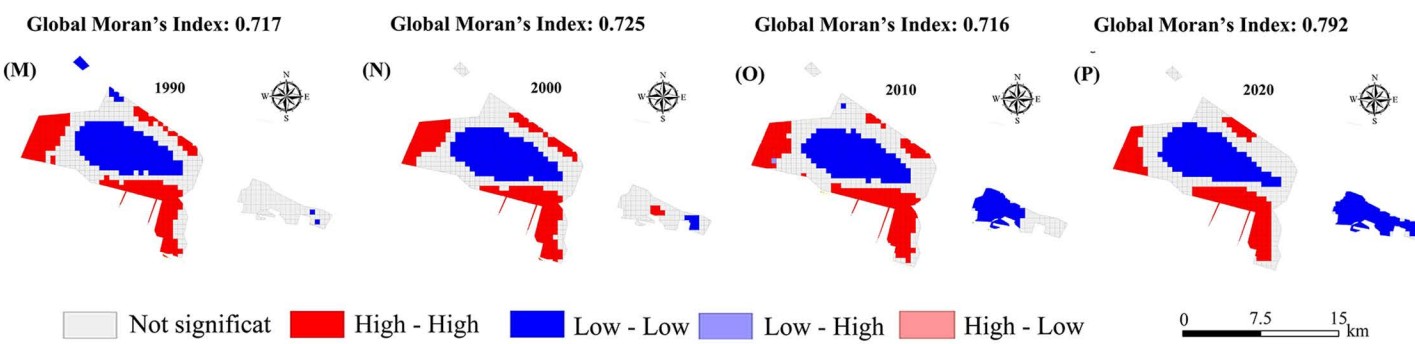

**Fig 5. Spatiotemporal distribution of ERI in oasis cities (1990—2020).**

(ranging from 0.127 to 0.144), indicating their fragility due to lower vegetation cover and soil fertility. These areas are generally characterized by extreme aridity and limited vegetation growth [41].

Table 6 lists the ERI threshold between different zones. These thresholds are critical reference points that indicate significant shifts in ecological risk between different landscape boundaries. They reflect the transition in landscape types and

the inherent ecological sensitivity and vulnerability of the zone to natural and anthropogenic pressures. The oasis-transition zone threshold (ranging from 0.08 to 0.085) decreases over time, which is likely due to improvements in management practices, such as soil conservation, vegetation restoration, and sustainable water management. In contrast, the desert-transition zone threshold (ranging from 0.111 to 0.118) shows minimal fluctuations, indicating a stable ERI and a lack of improvement despite management efforts.

## Spatiotemporal factors affecting the ERI

The observed and predicted $R^2$ values of the validation set of ERI were 0.73, 0.71, and 0.77 in 2000, 2010, and 2020, respectively, indicating that the RF regression model has high predictive ability. The RMSE values ranged from 0.0136 to 0.0165, and the RPD values ranged from 1.79 and 2.15, demonstrating the model's high stability (Fig 7a). The distances to towns and roads had the most significant influence on the ERI in 2000 (relative influences of 0.21 and 0.17, respectively), whereas the slope and elevation had the smallest influences (Fig 7b). The socioeconomic factors remained dominant from 2000 to 2010, but the relative importance of natural factors, such as precipitation and temperature, declined over time. The results for 2020 confirm the dominance of socioeconomic factors and the continued decline in the influence of natural factors on the ERI.

## Projections of trends in ERI and OLS

**Projections of trends in land-use change.** The actual and projected land use is shown in Fig 8a. The difference between the actual and simulated land use types in 2020 was lower than 3.5%. Cropland accounted for 59.3% of the

**Table 6. ERI Thresholds Between Adjacent Zones.**

| Year | 1990 | 2000 | 2010 | 2020 |
|---|---|---|---|---|
| Oasis—Transition zone | 0.084 | 0.085 | 0.082 | 0.08 |
| Transition zone—Desert | 0.116 | 0.118 | 0.114 | 0.111 |

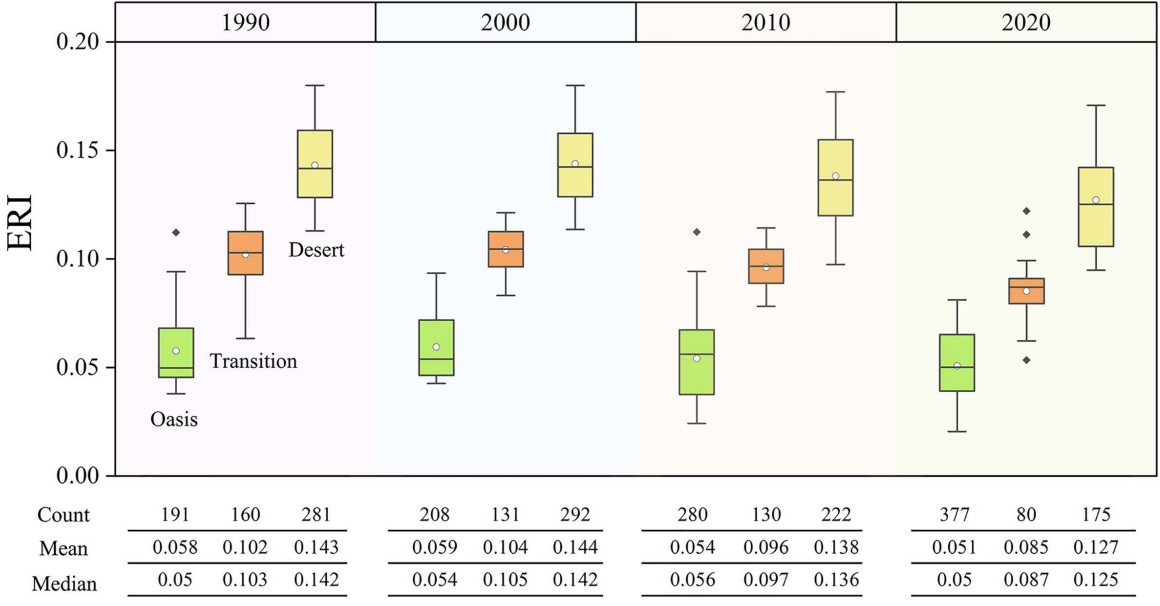

**Fig 6. Boxplots of the Ecological Risk Index (ERI) in the Oasis, Transition, and Desert Zones (1990–2020).**

total area of the simulated land use types, and unutilized land and grassland with medium to low cover accounted for 32.3%. The simulated and actual data had similar proportions of grassland, built-up land, and unutilized land. The area of cultivated land increased by 62.3% and 59.7% in the natural growth and government control scenarios, respectively, from 2020 to 2030, whereas the areas of low-cover and medium-cover grassland decreased. Forest land and high-cover grassland showed an upward trend, and the proportion of forest land and grassland increased in the government control scenario. The prediction results show that effective government management has limited the expansion of cultivated land and improved ecological stability.

**Prediction of trends in the ERI.** The projected spatial pattern of the ERI in 2030 is shown in Fig 8b. Under the natural growth scenario, the low-risk area of ERI is concentrated in the center of the study area, accounting for 36.3% of the area. The high-risk areas account for 4.7% of the area and are primarily located in the southern part of the study area. In contrast, the area proportion of low-risk areas increases to 39.6%, and the area share of high-risk areas decreases to 4.5% under the government control scenario. In this scenario, more low-risk areas occur, and the risk of the oasis ecosystem is significantly reduced.

**Prediction of trends in the OLS.** The proportion of land use types in oasis, transition, and desert zones in 2020 is listed in Table 7. Based on the trend of the simulated OLS components in 2030 (Fig 8c), the oasis area increased from 2020 to 2030 under natural and controlled scenarios, reaching 382.6 km$^2$ and 375.3 km$^2$, respectively. The transition zone area increased significantly under the government-controlled scenario, reaching 82.5 km$^2$, much higher than that under the natural growth scenario, indicating that government interventions improved the ecological restoration of oasis margins. The desert area decreased under both scenarios from 2020 to 2030, indicating the importance of conservation for biodiversity and limiting land expansion.

**a. 2000, 2010, 2020 ERI verification set based on RF regression.**

**b. 2000, 2010, 2020 ERI impact factor contribution based on RF regression.**

**Fig 7. Results of RF regression model validation and significance of influencing factors (desert area change and spatiotemporal characteristics of ERI).**

**a. trends in the evolution of land use structure in oasis city, 2020-2030.**

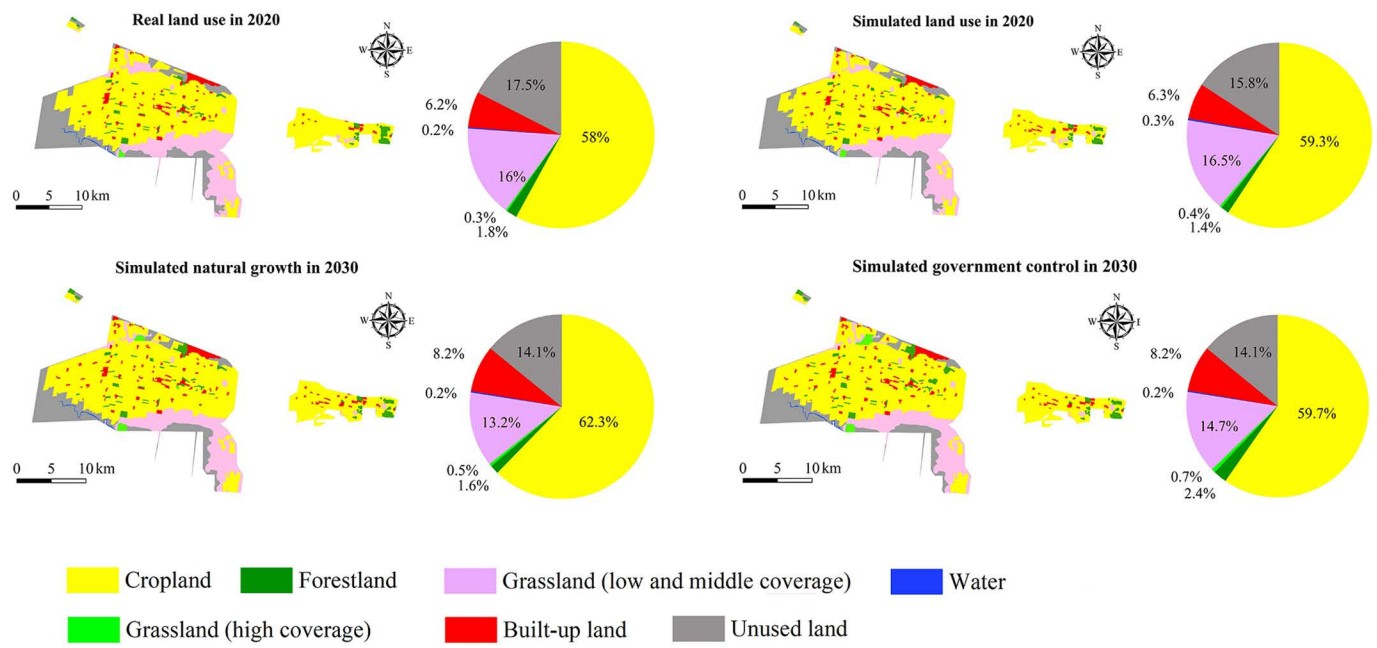

**b. spatial patterns of ERI in oasis cities under natural growth and government control scenarios in 2030.**

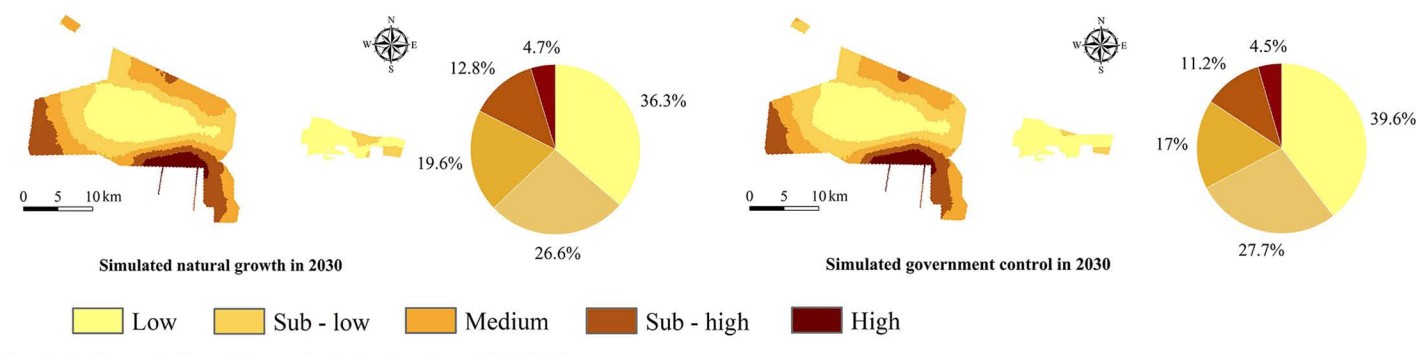

**c. trends in the evolution of the oasis circle structure, 2020-2030.**

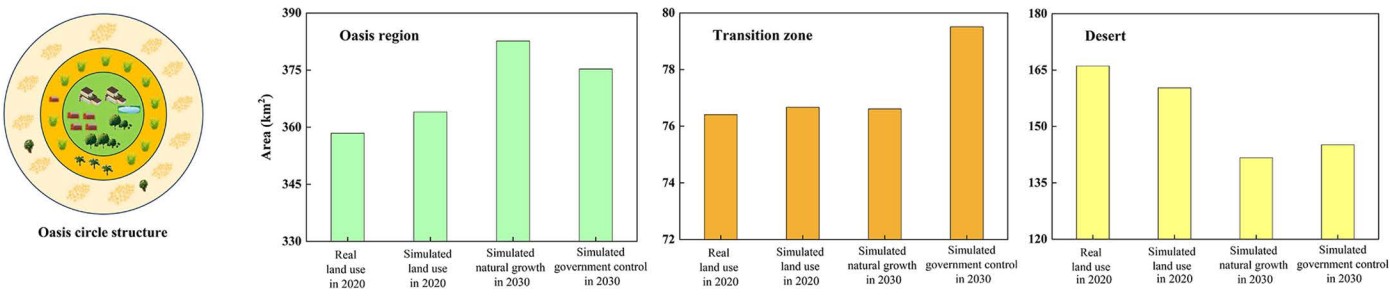

**Fig 8. Projected land use, ERI and OLS in 2030 under different conditions.**

## Discussion

### Potential link between OLS and ERI

The ERIs of different OLS regions were significantly different, suggesting a correlation between ERI and OLS (Fig 6). The average ERIs of oasis, transition, and desert zones were 0.05–0.06, 0.085–0.102, and 0.127–0.144 in 1990 to 2020, respectively, indicating an increasing trend. This result is similar to the findings of Wang et al., who concluded that the ecological risks ranged from low to high in core areas (where human activities were concentrated), buffer zones, and native areas (where anthropogenic disturbances were low) [42]. This result is likely due to differences in regional ecological management, the impacts of environmental disturbances, and differences in ecosystem resilience. Wang et al. and Liu and Tang concluded that differences in ecological management methods and the degree of disturbance affected ecological risk. They observed an increase in the oasis area as management measures improved [43,44]. Oasis zones have lower ecological risks due to higher vegetation cover and more scientific management. In contrast, transition and desert zones are more vulnerable to the negative impacts of climate change and human activities due to sparse vegetation and inadequate management [45,46]. In addition, the oasis area has a more robust ecosystem and is more resilient than the desert and transition zones. The concentration of human activities and inputs has reduced ecological risks in the oasis area. These factors contribute to the significant differences in ecological risk levels in different regions [47]. In addition, OLS and ERI show similar spatial patterns, i.e., a circular structure reflecting the spatial characteristics of the oasis zone (low ERI), transition zone (medium ERI), and desert zone (high ERI) (Fig 8c). Zhang et al. also observed a circular landscape pattern in the oasis region and significant differences in the landscape structure, which were related to the human activities and ecosystem resilience [48].

### Drivers of the ERI spatial pattern

The spatiotemporal evolution of the ERI exhibits significant heterogeneity, which, combined with the results of the empirical analysis, is influenced by natural and socioeconomic factors (Fig 7). We analyzed the dynamic evolution of regional ecological risk affected by natural and socioeconomic factors at different temporal scales. First, the influence of topographic relief (e.g., slope and elevation) was more stable (the combined relative influence was below 0.1). This factor affects soil erosion, contributing to the distribution of the ERI. Gan et al. and Chen et al. also observed small effects of elevation and slope on ecological risk [34,49]. Second, the effects of socioeconomic factors, such as infrastructure expansion, population density, and GDP growth, on the ERI became more dominant over time. The findings of Mansour et al. are similar to ours, with distance to roads, railroads (infrastructure), and population density as the main drivers of ecological change in the city [50]. Human activities alter ecosystem stability through land use change and environmental modifications. Thus, the ERI decreased from 0.041–0.182 (2000) to 0.018–0.171 (2020) (Fig 5). The expansion of roads and towns during urbanization significantly reduced the ERI [51]. The relative importance of climatic factors, such as precipitation and temperature, declined over the past decades, which may be related to improved infrastructure and regional adaptive management measures [52].

**Table 7. Proportion of land use types in oasis, transition, and desert zones in 2020.**

| Category | | Oasis | Transition zone | Desert |
|---|---|---|---|---|
| Percentage/% | Cropland | 88.0 | 12.0 | 0.0 |
| | Forestland | 85.8 | 13.2 | 0.9 |
| | Grassland (high coverage) | 59.4 | 9.9 | 29.7 |
| | Grassland (low and middle) | 11.8 | 20.4 | 67.7 |
| | Water | 100.0 | 0.0 | 0.0 |
| | Built-up land | 100.0 | 0.0 | 0.0 |
| | Unused land | 0.0 | 10.1 | 89.9 |

## RF model limitations and assumptions

Although the RF model performs well for handling nonlinear relationships and describing variable importance, it exhibited several limitations. Its results depend largely on the quality and accuracy of input data, such as socioeconomic indicators obtained from statistical yearbooks, which may have insufficient temporal resolution or consistency. Additionally, the model is a black box, making it difficult to interpret causal relationships and limiting its applicability to policy-making. Furthermore, the RF does not account for spatial autocorrelation, which may result in incorrect predictions in regions with significant spatial heterogeneity, such as the diverse landscapes of Xinjiang [53].

The RF model assumes that input data like GDP and population density are stable and representative during the study period. However, these variables are often subject to rapid changes due to economic shifts, policy updates, or migration, which static datasets may not capture. Although the RF model can handle multicollinearity, it assumes that variable interactions represent broader systemic relationships, which may overlook the influence of unobserved or hidden variables on ecological risks [54].

Future studies could consider integrating more dynamic and high-resolution socioeconomic datasets, such as those derived from satellite imagery or real-time urban monitoring systems. Additionally, coupling the RF model with spatial econometric methods or geographically weighted regression (GWR) could help account for spatial heterogeneity and improve the robustness of ecological risk predictions [55].

## Impacts of land use change on OLS and ecological risk

Land use change directly affects the evolution of the OLS [56,57]. The expansion of cultivated and built-up land and the reduction of unutilized land increased the oasis zone from 175.5 km$^2$ in 1990 to 190.6 km$^2$ in 2000, increasing ecological risk. Human activities intensified during this period due to economic development, and forest land was cleared for agriculture and construction. The lack of effective ecological management increased the ecological vulnerability of the transition and desert zones [58]. Insufficient vegetation cover weakened the transition zone's buffering ability, reducing ecosystem stability in some areas and landscape connectivity. Zhou et al. demonstrated that prolonged drought resulted in a decline in vegetation vitality in the oasis, affecting the region's ecological stability. Effective ecological management stabilizes vegetation growth by regulating the water environment [59]. Our results were in agreement with this study. Optimizing land-use management improved the OLA significantly and reduced ecological risk from 2000 to 2020. Due to ecological protection policies, more unutilized land was converted into oasis areas, and vegetation restoration and land management improved ecosystems, especially in transition zones and desert areas [60]. This change resulted in an expansion of the oasis (190.6 km$^2$ to 345.3 km$^2$), an increase in landscape connectivity, and a downward trend in the ERI from 0.182 to 0.171. Effective land use management slowed desertification and improved ecosystem resilience.

## Limitations and future research

This study simulated land use change using the PLUS model. Although this model has high predictive accuracy, it is limited by existing conditions. In addition, it may not fully capture unexpected changes in socioeconomic or environmental conditions [37,61].

We chose only one oasis desert city in Xinjiang, Tiemenguan City, for our study. This oasis desert city has undergone significant ecological changes and is not much affected by external variables, making it a representative case for revealing changes in the landscape structure of oasis desert cities and their driving mechanisms. The geographic and climatic conditions of Tiemenguan City make it an ideal study area. However, the uniqueness of this region may also limit the applicability of the conclusions to other arid and semi-arid regions. Oasis desert cities in different arid zones may have significant differences in climate, socioeconomic development level, land use, and management policies, resulting in different ecosystem responses and landscape structural changes.

Although this study provides an in-depth analysis of the conditions in Tiemenguan city, future research should be extended to other oasis desert cities to verify the generalizability of the findings. The mechanisms affecting ecological risks under different geographic, climatic, and socioeconomic conditions can be better understood by conducting cross-regional comparative studies in multiple regions [62,63]. We plan to conduct future studies in other cities in Inner Mongolia and Xinjiang and oasis desert cities in Central Asia to investigate ecosystem resilience under different conditions. Cross-city comparative studies will reveal differences in ecological risks in different regions and provide a more comprehensive theoretical basis for the ecological management of arid and semi-arid regions. Combining these research results with land management policies in different regions can provide more diverse references for future ecological management practices and help formulate targeted and regionally adapted sustainable development strategies.

This study highlights the critical role of urban ecological sustainability in addressing challenges faced by oasis desert cities like Tiemenguan city. Rapid urbanization and economic development in these regions exacerbate ecological risks, such as water scarcity, land degradation, and biodiversity loss, posing significant challenges to environmental stability and regional growth. This research emphasizes sustainable land-use practices and ecological restoration and provides scientific guidance for promoting balanced development. Targeted interventions, such as afforestation, land rehabilitation, and water resource management, are essential for restoring ecosystem functions and enhancing urban resilience. Ecological goals must align with economic development and ensure sustainable development strategies. By integrating ecological risk assessment, urban land use categories, and landscape characteristics, this study aims to provide scientific guidance to achieve urban sustainable development in arid regions under increasing environmental pressures.

## Conclusions

This study examined the spatiotemporal evolution characteristics of the OLS and ERI from 1990 to 2020 and their potential correlation. We used RF regression models to analyze the drivers of spatial changes in the ERI and the PLUS model to predict the trends of land-use change and the spatial characteristics of ERI and OLS in 2030 under natural development and government control scenarios. The following key insights were derived:

(1) The oasis zone expanded significantly during the 30-year study period, increasing from 175.5 km²in 1990 to 345.3 km²in 2020, whereas the desert zone shrank by 37.9% from 257.8 km² to 160.0 km². The ERI thresholds were 0.08–0.085 for the oasis-transition zone and 0.111–0.118 for the transition-desert zone, highlighting the transitional zone's critical role in ecological stability.

(2) Socioeconomic factors had a dominant influence on the ERI, with infrastructure expansion and population density exerting the most significant effects (the relative importance values exceeded 0.21). In contrast, the influence of natural factors like precipitation and temperature decreased over time. The highest ERI decrease was from 0.182 in 2000 to 0.171 in 2020, reflecting effective ecological management measures.

(3) Under the government-controlled scenario, ecological risk is projected to decline by 2030, with low-risk areas increasing from 36.3% under the natural growth scenario to 39.6% and high-risk areas decreasing from 4.7% to 4.5%. The transition zone area is expected to grow significantly to 82.5 km², whereas the oasis zone is predicted to expand to 375.3 km², a slower growth compared to the natural growth scenario (382.6 km²), demonstrating the efficacy of sustainable land-use policies.

This research underscores the importance of balancing socioeconomic development with ecological sustainability in arid regions. The findings highlight the need for integrated management approaches that enhance the resilience of oasis desert landscapes. Future studies should perform similar studies in diverse geographical and socioeconomic settings to promote sustainable urbanization in fragile ecosystems.

## Supporting information

**S1 Fig. Flowchart for extracting the oasis, desert, and transition zones.**
(TIF)

**S2 Fig. Random Forest structure with cross-validation.**
(TIF)

## Acknowledgments

We thank the School of Water Resources and Architectural Engineering, Shihezi University, and the Hydrology and Water Resources Management Center of the Second Division of the Xinjiang Production and Construction Corps.

## Author contributions

**Conceptualization:** Mingyue Sun, Hongguang Liu, Huan Cao.

**Data curation:** Mingyue Sun, Hongguang Liu, Rui Fang.

**Formal analysis:** Mingyue Sun, Yingsheng Dang, Ping Gong.

**Funding acquisition:** Hongguang Liu.

**Investigation:** Mingyue Sun, Yingsheng Dang, Pengfei Li, Rui Fang, Huan Cao.

**Methodology:** Hongguang Liu, Yingsheng Dang, Ping Gong.

**Project administration:** Hongguang Liu, Yingsheng Dang.

**Resources:** Hongguang Liu, Ping Gong.

**Software:** Mingyue Sun, Pengfei Li, Rui Fang, Huan Cao, Xiang Li, Hanji Xia, Fuhai Ye, Yong Guo.

**Supervision:** Yingsheng Dang, Ping Gong, Pengfei Li.

**Validation:** Mingyue Sun, Yingsheng Dang, Pengfei Li, Rui Fang, Xiang Li, Hanji Xia, Fuhai Ye, Yong Guo.

**Visualization:** Mingyue Sun, Ping Gong, Pengfei Li, Rui Fang, Huan Cao, Xiang Li, Hanji Xia, Fuhai Ye, Yong Guo.

**Writing – original draft:** Mingyue Sun.

**Writing – review & editing:** Mingyue Sun, Hongguang Liu, Yingsheng Dang.

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
