## [Decision Letter · Decision Letter 0]

22 Dec 2024

PONE-D-24-50511Landscape Structure Evolution and Ecological Risk Evaluation of Oasis Desert Cities: A Case study of Tiemenguan CityPLOS ONE

Dear Dr. Hongguang,

Thank you for submitting your manuscript to PLOS ONE. After careful consideration, we feel that it has merit but does not fully meet PLOS ONE’s publication criteria as it currently stands. Therefore, we invite you to submit a revised version of the manuscript that addresses the points raised during the review process.

We look forward to receiving your revised manuscript.

Kind regards,

Pradeep Kumar Badapalli

Academic Editor

PLOS ONE

Journal Requirements:

[This research study was supported by both Divisional and Municipal Science and Technology Tackling Program Projects (2024GG2301 and 2023GG1502)].

6. We note that Figures 2, 3, 4, 5 and 8 in your submission contain [map/satellite] images which may be copyrighted. All PLOS content is published under the Creative Commons Attribution License (CC BY 4.0), which means that the manuscript, images, and Supporting Information files will be freely available online, and any third party is permitted to access, download, copy, distribute, and use these materials in any way, even commercially, with proper attribution. For these reasons, we cannot publish previously copyrighted maps or satellite images created using proprietary data, such as Google software (Google Maps, Street View, and Earth). For more information, see our copyright guidelines: http://journals.plos.org/plosone/s/licenses-and-copyright.

1. You may seek permission from the original copyright holder of Figures 2, 3, 4, 5 to publish the content specifically under the CC BY 4.0 license.

Additional Editor Comments:

Dear Dr. Hongguang,

Thank you for submitting your manuscript.

After reviewing the feedback from our reviewers, we are pleased to inform you that your manuscript requires "minor revisions" before it can be accepted for publication.

Please address the reviewers’ comments carefully and submit the revised manuscript.

Ensure that your response clearly outlines the changes made.

We look forward to your revised submission.

Reviewers' comments:

Reviewer's Responses to Questions

**Comments to the Author**

1. Is the manuscript technically sound, and do the data support the conclusions?

Reviewer #1: Yes

Reviewer #2: Yes

2. Has the statistical analysis been performed appropriately and rigorously? 

Reviewer #1: Yes

Reviewer #2: Yes

3. Have the authors made all data underlying the findings in their manuscript fully available?

Reviewer #1: Yes

Reviewer #2: Yes

4. Is the manuscript presented in an intelligible fashion and written in standard English?

Reviewer #1: Yes

Reviewer #2: Yes

5. Review Comments to the Author

Reviewer #1: Peer Review of Manuscript ‘Landscape Structure Evolution and Ecological Risk Evaluation of Oasis Desert Cities: A Case Study of Tiemenguan City’

1. Summary of the Research

This study investigates the spatiotemporal evolution of oasis landscape structures (OLS) and ecological risk indices (ERI) in Tiemenguan City, a typical oasis desert city in northwestern China, from 1990 to 2020. The research focuses on the relationship between OLS and ERI, the drivers of ecological risk, and projections under natural and government-regulated scenarios using models like Random Forest (RF) and PLUS. The main findings indicate continuous oasis expansion, decreasing ERI, and the dominance of socioeconomic factors in driving ecological risks. The study provides valuable insights into sustainable development in arid zones.

Strengths

• Comprehensive integration of remote sensing, modelling techniques, and ecological indices.

• Detailed methodology ensuring replicability and reliability.

• Novel insights into the dynamics of oasis desert cities under varying scenarios.

Weaknesses

• Limited generalizability due to the study's focus on a single city.

• Lack of clarity in certain figures and tables, which could hinder understanding.

• The discussion of limitations and future directions could be expanded.

Recommendation: Accept with minor revisions.

2. Examples and Evidence

Major Issues

1. Clarity in Methodology: While the methods are detailed, some equations (e.g., ERI calculation) lack clear explanations of parameters. Clarify how weights (a, b, c) were chosen and the rationale for using specific indices.

2. Generalizability: The focus on Tiemenguan City limits applicability to other arid zones. Future research should include comparative studies across multiple cities.

3. Figures and Tables: Certain visualizations (e.g., Fig. 6, Table 6) lack sufficient captions and are not self-explanatory. Ensure all visuals are clear and provide standalone explanations.

4. Scenarios: The assumptions for natural and government-controlled scenarios need better justification, including more details about specific policy interventions.

Minor Issues

1. Grammatical inconsistencies, particularly in the introduction and conclusion sections (e.g., sentence fragments in lines 90–92).

2. Enhance the discussion of RF model limitations and assumptions, particularly regarding socioeconomic data sources.

3. Standardize references to maintain consistency with journal formatting guidelines (e.g., line 477–501).

4. Expand the ethical implications of the study, especially given its focus on urban ecological sustainability.

Reviewer #2: The abstract is correct, but more information about how the parameters were done could be included.

The information in the "Introduction" section is also complete, but linking words may be applied to join different ideas. Figures should be increased in size as they are difficult to read. In line 269, delete "of" in "Area statistics for OF oases..."

Line 414, delete ".." in the phrase: "the evolution of the OLS [50, 51]. ."

In the "Discussion" section, it would be appropriate to relate results and discussions directly with previous related studies.

The "Conclusion" section needs to be improved. It is closer to a synthesis of the results than to conclusions.

Some of the references are in incorrect format, mainly in journal names.

6. PLOS authors have the option to publish the peer review history of their article (what does this mean? ). If published, this will include your full peer review and any attached files.

**Do you want your identity to be public for this peer review?** For information about this choice, including consent withdrawal, please see our Privacy Policy .

Reviewer #1: **Yes: ** Mastawal Melese

Reviewer #2: No

---

## [Author Response · Author response to Decision Letter 1]

15 Jan 2025

Thanks to the editors and reviewers, my “revisions” have been uploaded to the system as files named “Cover letter”, “Response to Reviewers 1”, ‘Response to Reviewers 2’, and ‘Revised Manuscript with Track Changes’. and “Manuscript”.

---

## [Decision Letter · Decision Letter 1]

25 Feb 2025

PONE-D-24-50511R1Landscape Structure Evolution and Ecological Risk Evaluation of Oasis Desert Cities: A Case study of Tiemenguan CityPLOS ONE

Dear Dr. Hongguang,

Thank you for submitting your manuscript to PLOS ONE. After careful consideration, we feel that it has merit but does not fully meet PLOS ONE’s publication criteria as it currently stands. Therefore, we invite you to submit a revised version of the manuscript that addresses the points raised during the review process.

We look forward to receiving your revised manuscript.

Kind regards,

Pradeep Kumar Badapalli

Academic Editor

PLOS ONE

Journal Requirements:

Additional Editor Comments:

Dear Author,

Based on the reviewers' reports, your manuscript has been assigned a Minor Revision. You are invited to revise the manuscript accordingly and resubmit it for reconsideration. Please ensure that all reviewer comments are carefully addressed to improve the quality of your work.

I look forward to the possibility of acceptance upon revision.

We look forward to receiving your revised submission.

Reviewers' comments:

Reviewer's Responses to Questions

**Comments to the Author**

1. If the authors have adequately addressed your comments raised in a previous round of review and you feel that this manuscript is now acceptable for publication, you may indicate that here to bypass the “Comments to the Author” section, enter your conflict of interest statement in the “Confidential to Editor” section, and submit your "Accept" recommendation.

Reviewer #1: All comments have been addressed

Reviewer #2: All comments have been addressed

2. Is the manuscript technically sound, and do the data support the conclusions?

Reviewer #1: Yes

Reviewer #2: (No Response)

3. Has the statistical analysis been performed appropriately and rigorously? 

Reviewer #1: Yes

Reviewer #2: (No Response)

4. Have the authors made all data underlying the findings in their manuscript fully available?

Reviewer #1: No

Reviewer #2: (No Response)

5. Is the manuscript presented in an intelligible fashion and written in standard English?

Reviewer #1: Yes

Reviewer #2: (No Response)

6. Review Comments to the Author

Reviewer #1: The authors have addressed the comments comprehensively and systematically. Here is the evaluation of their revisions

A. Major Issues Addressed

1. Clarity in Methodology

The authors added detailed explanations of the ERI calculation parameters and weights in the revised text (Table 3 and accompanying paragraphs). The rationale for selecting weights (a, b, c) based on ecosystem stability and land use types is now clearly articulated.

Verification: Satisfactorily addressed.

2. Generalizability

The "Limitations and Future Research" section now explicitly acknowledges the study's geographic specificity and proposes future comparative studies in other arid regions (e.g., Inner Mongolia, Central Asia).

Verification: Adequately addressed with plans for broader applicability.

3. Figures and Tables

They revised the captions for Figure 6 and Table 6, making them self-explanatory. They provided detailed explanations of the ecological risk index (ERI) and how it varies across different zones.

Verification: Likely addressed, pending final figure checks by the editor.

4. Scenario Assumptions

The government-controlled scenario now references specific policies (e.g., "Green Development Policy," "14th Five-Year Plan") to justify assumptions. This strengthens the scenario’s validity.

Verification: Fully addressed.

B. Minor Issues Addressed

1. Grammatical Inconsistencies

The authors have corrected grammatical issues in the introduction (lines 90–92 revised to lines 104–108). They also performed a language check throughout the paper.

Verification: Likely resolved.

2. RF Model Limitations

The Discussion now elaborates on RF limitations (e.g., data dependency, spatial autocorrelation) and suggests integrating spatial econometric methods.

Verification: Thoroughly addressed.

3. Reference Standardization

References in the revised manuscript follow a consistent format (e.g., journal abbreviations, author order).

Verification: Confirmed in the provided references section.

4. Ethical Implications

They added a discussion on the ethical considerations of urban ecological sustainability in the revised version.

Verification: Addressed; sufficient for the scope.

Conclusion

The authors have thoroughly addressed the requested revisions. The manuscript now provides clearer explanations of methodology, better justification for its assumptions, improved visual clarity, and an expanded discussion on limitations and future directions. Based on this, the manuscript appears ready for acceptance.

Recommendation: Accept

Reviewer #2: (No Response)

7. PLOS authors have the option to publish the peer review history of their article (what does this mean? ). If published, this will include your full peer review and any attached files.

**Do you want your identity to be public for this peer review?** For information about this choice, including consent withdrawal, please see our Privacy Policy .

Reviewer #1: **Yes: ** Mastawal Melese

Reviewer #2: No

---

## [Author Response · Author response to Decision Letter 2]

4 Mar 2025

We greatly appreciate the editor and reviewers for your constructive and positive comments and suggestions on our manuscript.

We have carefully studied the reviewers’ comments, made revisions to the revised manuscript, and did our best to improve our manuscript according to the comments.

We have carefully addressed all comments raised during the initial review process and resubmitted the revised manuscript accordingly. In the latest decision email, we noted that Reviewer 1 has recommended "Accept" and confirmed that all concerns were resolved, while Reviewer 2 did not submit additional comments in this round. To ensure full compliance with journal requirements, we have also thoroughly checked the manuscript format against the Author Guidelines and confirmed its adherence.

We deeply value the time and expertise invested by the editorial team and reviewers in improving our work. Please let us know if further clarifications or adjustments are needed. We look forward to your decision and remain available for any follow-up requests.

Thank you for your ongoing support.

---

## [Editor Report · Decision Letter 2]

11 Mar 2025

Landscape Structure Evolution and Ecological Risk Evaluation of Oasis Desert Cities: A Case study of Tiemenguan City

PONE-D-24-50511R2

Dear Dr. Hongguang,

We’re pleased to inform you that your manuscript has been judged scientifically suitable for publication and will be formally accepted for publication once it meets all outstanding technical requirements.

Kind regards,

Pradeep Kumar Badapalli

Academic Editor

PLOS ONE

Additional Editor Comments (optional):

Dear Authors,

I am pleased to inform you that your manuscript has been accepted for publication. The reviewers are satisfied with your responses and the revisions made to address their comments. Congratulations on your successful submission!

Best regards

Dr. Pradeep Kumar Badapalli
---

## [Editor Report · Acceptance letter]

PONE-D-24-50511R2

PLOS ONE

Dear Dr. Liu,

I'm pleased to inform you that your manuscript has been deemed suitable for publication in PLOS ONE. Congratulations! Your manuscript is now being handed over to our production team.

Kind regards,

on behalf of

Dr. Pradeep Kumar Badapalli

Academic Editor

PLOS ONE